# Time-Myopic Go-Explore: Learning A State Representation for the Go-Explore Paradigm

## Abstract

Very large state spaces with a sparse reward signal are difficult to explore. The lack of a sophisticated guidance results in a poor performance for numerous reinforcement learning algorithms. In these cases, the commonly used random exploration is often not helpful. The literature shows that this kind of environments require enormous efforts to systematically explore large chunks of the state space. Learned state representations can help here to improve the search by providing semantic context and build a structure on top of the raw observations. In this work we introduce a novel time-myopic state representation that clusters temporally close states together while providing a time prediction capability between them. By adapting this model to the Go-Explore paradigm (Ecoffet et al., 2021b), we demonstrate the first learned state representation that reliably estimates novelty instead of using the hand-crafted representation heuristic. Our method shows an improved solution for the detachment problem which still remains an issue at the Go-Explore Exploration Phase. We provide evidence that our proposed method covers the entire state space with respect to all possible time trajectories — without causing disadvantageous conflict-overlaps in the cell archive. Analogous to native Go-Explore, our approach is evaluated on the hard exploration environments MontezumaRevenge, Gravitar and Frostbite (Atari) in order to validate its capabilities on difficult tasks. Our experiments show that time-myopic Go-Explore is an effective alternative for the domain-engineered heuristic while also being more general. The source code of the method is available on GitHub: `made.public.after.acceptance`.

**Keywords:** Exploration, Self-Supervised Learning, Go-Explore

## 1    Introduction

In recent years, the problem of sufficient and reliable exploration remains an area of research in the domain of reinforcement learning. In this effort, an agent seeks to maximize its extrinsic discounted sum of rewards without ending up with a sub-optimal behavior policy. A good exploration mechanism should encourage the agent to seek novelty and dismiss quick rewards for a healthy amount of time to evaluate long-term consequences of the action selection.

Four main issues are involved when performing an exploration in a given environment: (i) catastrophic forgetting (Goodfellow et al., 2013) as the data distribution shifts because the policy changes, (ii) a neural network's overconfident evaluation of unseen states (Zhang et al., 2018), (iii) sparse reward Markov Decision Processes and (iv) the exploration-exploitation trade-off (Sutton and Barto, 2018; Ecoffet et al., 2021a). The latter causes a significant problem: the greater the exploitation the less exploration is done. Hereby, making novelty search less important when the agent can easily reach states with a large rewards.

To address these difficulties, Ecoffet et al. (2021b) propose a new approach called Go-Explore and achieve state-of-the-art results on hard exploration problems. However, Go-Explore relies on hand-crafted heuristics. In our work we replace their discrete state representations with learned representations particularly designed to estimate elapsed time to improve the novelty estimation. The time distance between two states is used to build an abstraction level on top of the raw observations by grouping temporally close states. Equipped with this capability, our model can decide about the acceptance or rejection of states for the archive memory and maintain additionally exploration statistics.

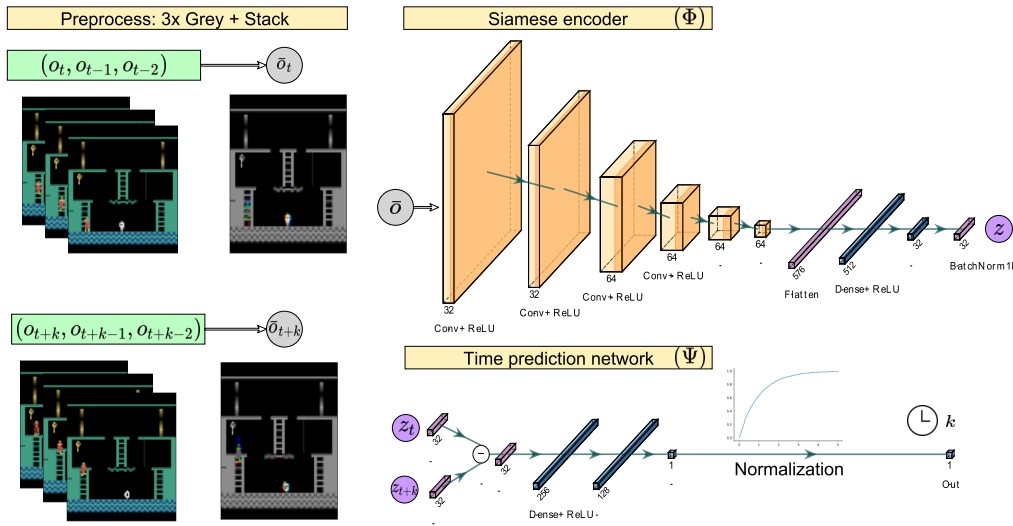

Figure 1: Model architecture. For details see Section 3.

**Contribution**. This work attempts to improve the exploration problem by introducing a learnable novelty estimation method that is applicable to arbitrary inputs. In the experiment section, we demonstrate the reliability of our method by comparing its results with native Go-Explore and more broadly with several other exploration-based and baseline approaches. The main contributions of this paper are the following:

1. We introduce a new *novelty estimation* method consisting of a siamese encoder and a time-myopic prediction network which learns problem-specific representations that are useful for time prediction. The time distances are later used to determine novelty which generate a time-dependent state abstraction.

2. We implement a new *archive* for Go-Explore that includes a new insertion criterion, a novel strategy to count cell visits and a new selection mechanism for cell restarts.

## 2 PROBLEMS OF GO-EXPLORE

Go-Explore maintains an archive of saved states that are used as milestones to be able to restart from intermediate states, hereby prevent detachment and derailment (as discussed in Ecoffet et al. (2021a)). It generates its state representation by down-scaling and removing color (see Figure 2 left). The exact representation depends on three hyperparameters (width, height, pixel-depth), which have to be tuned for each environment. If two distinct observations generate the same encoding they are considered similar and dissimilar otherwise. Thus, all possible states are grouped into a fixed number of representatives, which leads to overlap-conflicts when two distinct observations receive the same encoding. In these cases one of the states has to be abandoned, which might be the reason that for the Atari environment Montezuma's Revenge only 57 out of 100 runs reached the second level in the game (as reported in Ecoffet et al. (2021a)). We conjecture that their replacement criterion favors a certain state over another, so the exploration will be stopped at the abandoned state. This can result in decoupling an entire state subspace from the agent's exploration endeavor.

A related issue emerges by ignoring the spatio-temporal semantics between states that are grouped together into a single representation. The down-scaling method is just compressing the image information without considering its semantic content. The consequence is a replacement criterion that resolves archive conflicts between states that are neither spatially nor temporally in a close relationship. Therefore a conflict solver has to resolve illogical and non-intuitive conflicts, because in these cases it is not obvious which state should be favored. We have no information about which potentially reachable states are more promising. On top of that, the cell representation is only suitable for states represented by small images. If we have images with higher resolutions the representations

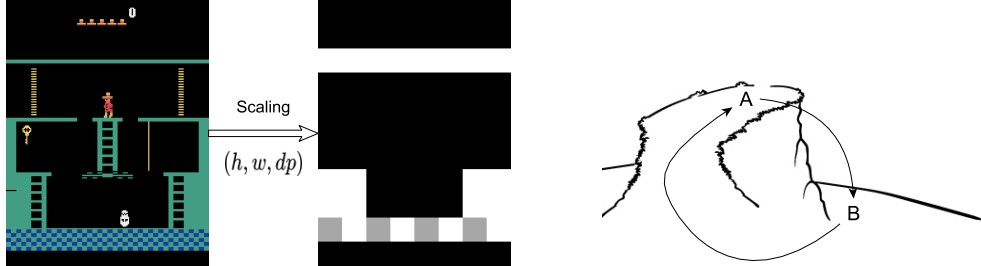

Figure 2: (left panel) Go-Explore down-scaling operation with hyperparameters $(h, w, dp)$. (right panel): time distances are directed. Jumping off the cliff is faster, than hiking up (image from IHC).

could get either abstract (and hereby limiting the exploration progress) or too narrow resulting in an overflowing archive. Moreover and from a general point of view, it is not clear how we could usefully down-scale an input observation that consists of partial or no image information at all.

## 3 LEARNING TIME-MYOPIC STATE REPRESENTATIONS

The basic idea of Go-Explore is to save intermediate results, so that the environment state can be restored from a memory to try new action sequences and obtain different outcomes. However, the difficulty is to decide when to store a state for returning. Go-Explore solves this by a heuristic that lead to some non-intuitive decisions when dealing with representation conflicts as discussed in the previous section.

In the following we show how state representations can be learned that are particularly designed to predict the time progress. Equipping Go-Explore with this model will avoid archive conflicts and gives a better control over the novelty estimation problem. We call our method *time-myopic* since it is trained to predict the elapsed time only up to a particular distance.

### 3.1 PREPROCESSING AND SIAMESE ENCODER

As a first step, we preprocess for every timestep $t$ a stack of the last three RGB observations $o_t, o_{t-1}, o_{t-2}$. For this we convert each observation into a single gray-scale frame to generate a compressed input format (see Fig. 1). The result $\bar{o}_t$ consist of three color channels where each channel represents one gray-scale observation. This format captures time-critical information and allows us to infer properties such as velocity and acceleration of objects.

Given a preprocessed observation $\bar{o}_t$, we apply a siamese convolutional neural network (CNN) $\Phi_\theta$ to obtain a latent vector encoding $z_t$

$$\Phi_\theta(\bar{o}_t) = z_t. \tag{1}$$

The exact architecture of the CNN $\Phi_\theta$ is shown at the top right in Figure 1. The goal is to learn a useful embedding that captures the information to accurately predict the elapsed time $k$ between two given observations $\bar{o}_t$ and $\bar{o}_{t+k}$. We choose siamese networks since they demonstrated good results (Koch, 2015; Ermolov and Sebe, 2020; Chen et al., 2021) in distinguishing deviations between distinct inputs or respectively capturing similarity.

### 3.2 TIME-PREDICTION NETWORK

After the encoding procedure, a pair of observation encodings $(z_t, z_{t+k})$ allows us to estimate the elapsed time between the elements of a pair. For this, we feed the difference between $z_t$ and $z_{t+k}$ into another fully-connected multi-layer neural network $f_\theta$ that outputs a real number. The whole time prediction network is written as $\Psi_\theta$ and includes a normalization (see bottom right Fig. 1)

$$\Psi_\theta(z_t, z_{t+k}) = 1 - e^{-\max(f_\theta(z_{t+k}-z_t), 0)} \qquad \in [0, 1]. \tag{2}$$

In Equation (2) the model predicts the time distance from the starting point $z_t$ to the destination $z_{t+k}$ where a swap can result in a different outcome. We obtain the myopic property by normalizing

the output with the function $g(x) = 1 - e^{-x}$ that allows us to specify a prediction range $[0, L]$ for the normalization interval $[0, 1]$. When we want to obtain the predicted time distance $k$ we need to multiply the normalized output with the upper bound $L$ and vice versa for converting a distance $k$ to the internal representation

$$k \approx L \cdot \Psi_\theta(z_t, z_{t+k}). \tag{3}$$

### 3.3 LOSS FUNCTION AND ITS IMPLICATIONS

The time prediction loss objective $L^{\text{time}}(\theta)$ minimizes the mean squared error between the predicted time distance and the true distance $k$

$$L^{\text{time}}(\theta) = \mathbb{E}_t \left[ \left( \Psi_\theta(z_t, z_{t+k}) - \min(k/L, 1) \right)^2 \right], \tag{4}$$

where the correct distance $k$ is known from the sampled trajectories after the environment interaction. The neural network computation includes the normalization $g(x)$ to prevent exploding gradients and introduces the upper time window bound $L$ that limits the network's expressivity beyond $L$ timesteps. This constraint simplifies the task, since correct forecasts of longer periods of time become unimportant (Makridakis et al., 2018). The myopic view is helpful to reliably estimate novelty, because the model does not need to handle a precise time measurement between states that are far away from each other. By considering only local distances, the predictions get easier and they are more reliable. So when the network encounters a state-pair with a large distance $k > L$, we do not care about the exact true distance. But instead, the model should indicate that the states are far away from each other by outputting the upper bound $L$.

### 3.4 TIME DISTANCE IS POLICY DEPENDENT AND DIRECTED

Each singular timestep represents a state transition on the underlying MDP where the time-related reachability is dependent on the current policy. This means that two distinct policies could reach a certain state with a different number of timesteps. Also, since time distances are directed we have to input our encodings $(z_A, z_B)$ in the right order into $\Psi_\theta$. For intuition, Figure 2 (right) shows a visual example where a state $A$ is on top of a small cliff and point $B$ is at the bottom of it. To reach $B$ from $A$ an agent could use the shortcut and jump down the cliff, but it might not be possible to jump back up. Therefore the agent needs to find a different path that might increase the elapsed time to reach $A$ from $B$. For that reason we assume that the distances of our time prediction function can yield different results for $\Psi_\theta(z_A, z_B)$ and $\Psi_\theta(z_B, z_A)$.

## 4 TIME-MYOPIC GO-EXPLORE

To integrate our learned representation model from the previous section into the native Go-Explore method, we have to make some adjustments, since our state encodings are continuous (and no longer discrete). This makes it harder to decided similarity between two distinct states. In the following we explain how this is achieved.

### 4.1 ARCHIVE INSERTION CRITERION

The time prediction capability of $\Psi_\theta$ allows us to propose a new archive criterion. Our criterion is an insertion-only method which adds a state to the archive when it is novel enough. In this way the detachment problem is entirely solved by not abandoning archive states. We initialize the archive by inserting the encoding $z_{c_1}$ of the environment's start state. This entry will begin the exploration effort by searching for novelty around the starting point. In order to do that the agent samples trajectories from the restored state $c_1$ where we collect new states that we call archive candidates. A new candidate encoding $z_K$ is evaluated with *every* archive entry $z_C$ by the time prediction function. The predicted value $\Psi_\theta(z_C, z_K)$ is thresholded by some hyperparameter $T_d$ to determine the acceptance or rejection of the current candidate. If the threshold is surpassed for *every* comparison, the candidate $K$ is added to the archive. Note, that we choose the time distance direction $\Psi_\theta(z_C, z_K)$ since we want to move away from the archive entries. When this happens, we conclude that the agent has reached a currently unknown part of the state space which has a sufficient novelty with respect to the archive-known subspace. In this way, the candidate evaluation gains a *global* view on the exploration progress instead of considering a local trajectory-based perspective (Badia et al., 2020; Savinov et al., 2018). A short example is shown in Table 1.

Table 1: Archive insertion criterion where the model's expressivity is in the time window $[0, L = 20]$. A new candidate $z_K$ is compared to all existing cells in the archive. If one cell is too close, the candidate is rejected.

| Comparison | | Time estimation | Evaluation | |
| Cell | Candidate | $\Psi_\theta(z_c, z_K)$ | $20 * T_d > 13$ | Insert? |
| --- | --- | --- | --- | --- |
| $z_{c_1}$ | $z_K$ | 0.3934 | 7.86 | no |
| $z_{c_2}$ | $z_K$ | 0.7568 | 15.13 | yes |
| $z_{c_3}$ | $z_K$ | 0.6321 | 12.64 | no |
| $z_{c_4}$ | $z_K$ | 0.9334 | 18.66 | yes |
| ... | $z_K$ | ... | ... | ... |

## 4.2 VISIT COUNTER

To ensure successful progression in the exploration, it is common practice to track the number of cell visits $C_{\text{visits}}$ for every cell $C$ in the archive. With respect to the return selection, the visit counter of a cell increases by one when it is selected as a starting point. Furthermore, the visit counter increments when the time distance to an archive candidate is lower than a visit threshold $T_v$. This evaluation runs simultaneously to the application of our insertion criterion where we compute all time distances to the candidate.

## 4.3 CELL SELECTION CRITERION

Native Go-Explore made the design choice to replace cell entries with small scores to states with larger scores. This significantly boosts the exploration by abandoning states which might have been sufficiently explored. Time-myopic Go-Explore does not have this ability, because the state representations are continuous and the encodings of two states with weak spatio-temporal semantics are far away from each other. Also, our method should not overwrite archive entries, because they could be inserted again with clean visits statistics. To bias exploration towards cells with larger scores, we calculate and combine the native selection weight $W_{\text{visit}}$ with our score weight $W_{\text{score}}$ that contains the reached cell score $C_{\text{score}}$ (sum of undiscounted cumulative reward). Both quantities ($C_{\text{visits}}, C_{\text{score}}$) are stored in the archive for every cell $C$.

$$W = \underbrace{\frac{1}{\sqrt{C_{\text{visits}} + 1}}}_{\text{visit weight } W_{\text{visit}}} \cdot \underbrace{\max\left(\frac{C_{\text{score}}}{\max_{C'} C'_{\text{score}} + 1}, \alpha\right)}_{\text{score weight } W_{\text{score}}}. \tag{5}$$

The outer maximum in $W_{\text{score}}$ ensures that states with lower scores are explored as well (with chosen hyperparameter $\alpha = 0.075$). The inner maximum normalizes the experienced scores between zero and one.

## 5 TECHNICAL DETAILS FOR TRAINING

In order to improve the model quality we explain next three additional training routines that enhance the time prediction of our model.

## 5.1 SIMILARITY AND DISSIMILARITY

Optimizing only on close-by pairs (e.g. with time distance $k \leq L$) will lead to wrong estimates for distant pairs (i.e. $k > L$), which might be the majority of cells. Thus we have to ensure that our dataset includes also dissimilar pairs where the observations are at least $L$ timesteps away from each other. The distance of a dissimilar pair is trained on the value that corresponds to the maximal time estimation $L$, which does not represent the actual distance. Instead it signals a sufficiently large temporal space between them. In this way, we try to find encodings such that the embedding region around a state only includes other encodings that are within the defined time window $[0, ..., L]$.

## 5.2 AVOIDING TEMPORAL AMBIGUITY

Suppose there are two observations $\bar{o}_{t+k_1}$ and $\bar{o}_{t+k_2}$ that happened at different points in time, but which are pixel-wise identical. To define a single distance to some other observation $\bar{o}_t$, we will use the minimum of $k_1$ and $k_2$. To quickly identify pixel-wise identical observations we are using the MD5 hash function (Rivest, 1992).

## 5.3 PROXY CELLS AND LOCAL DATASETS

Usually the time prediction model would be trained on trajectories that always start with an archive state. To extend the training data we additionally generate trajectories (only for training) that start from so-called proxy cell states that are temporally close to an archive cell. We collect these proxies by choosing randomly states in the time distance interval $[T_{\text{p-low}}, T_{\text{p-high}}]$ to their respective archive cell and replace them periodically.

To further increase variability of the dataset, we add small local datasets for each cell that add some data points within its proximity. Every dataset holds a few hundred pairs where we store the time distance between the cell state and a temporally close state. This is necessary because otherwise our network would forget about certain cells and their neighboring states since they are not visited anymore due to the selection weighting $W = W_{\text{visits}} * W_{\text{score}}$.

## 6 EXPERIMENTS

The experiment section covers the following topics: in Section 6.1 we study the encoder properties and assess how well the time can be predicted. In Section 6.2 we compare our approach, the original Go-Explore and other related methods. In Section 6.3 we analyze the archive for the native and our proposed time-myopic Go-Explore.

## 6.1 WHAT DOES OUR MODEL LEARN?

To demonstrate the properties of our model, we re-create an experiment from Ermolov and Sebe (2020) which is shown in Figure 3. A representation-learning network is trained on 400k observations from the Atari environment Montezuma's Revenge where the training data is gathered by a random actor at the environment's start state. After optimization, the network is asked to encode the observation sequence from the trajectory shown in Figure 3 (top-left). Most of the encodings are extrapolated, because a random policy is not able to reliably execute the action sequence that generated the observations. Therefore an extrapolation starts roughly between the checkpoint 3 and 4. In this experiment our model uses 9k observations which are sufficient to show a good result. The 32-dimensional encodings are projected with the t-SNE method into a two-dimensional space where it uncovers the interesting relationship between the data points. Temporally close states are grouped together and are strung on a thread while the sequence unfolds. The distance between two imminent states does not collapse and it provides useful semantics for time measurements. Remember that the chosen trajectory has no greater meaning for our model and it is perceived as arbitrary like any other possibly selected sequence. Subsequently, the rich structure is also present when performing PCA on the encodings (see Appendix Figure 5).

The time evaluation demonstrates good results where the network has training data. However, the extrapolation capability on time distances is limited. But surprisingly we can see that the encoder even places unseen states close to their temporal neighbors resulting in local semantic integrity. At some point the extrapolation of time distances becomes unreliable which will improve with more novel data for optimization. Once the network was trained on more data during a complete run, the prediction quality gets a lot better (see Appendix Figure 6).

## 6.2 COMPARISON ON HARD EXPLORATION ENVIRONMENTS

We evaluate our method on the hard exploration environments *Montezuma's Revenge, Gravitar, Frostbite* (Atari) and compare it with (i.) related methods (ii.) and native Go-Explore. We prepare our experiments by adjusting the natively used random action-repeating actor (Ecoffet et al., 2021a). Time-myopic Go-Explore decreases the actor's action repetition mean $\mu$ from 10 to 4. This is necessary since the native method does not care about learning a state representation while our learned model depends on a robust data collection for the time predictions. Native Go-Explore has

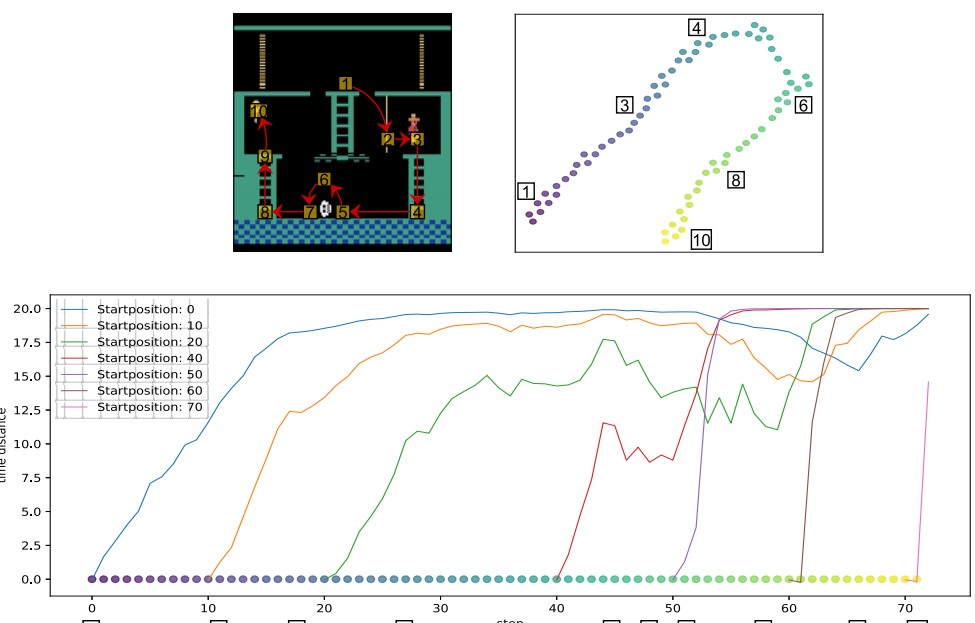

Figure 3: Visualization of the model performance. The top-left image is an observation sequence from the Atari environment Montezuma's Revenge, which is generated by a human demonstration. The top-right image is the t-SNE (van der Maaten and Hinton, 2008) projection of the observation encodings generated by the siamese CNN $\Phi_\theta$. The bottom image shows time estimates of our prediction network $\Psi_\theta$ for $L = 20$. This model has the ability to calculate the time distance for a pair of observations. E.g. we can select several encodings $z_0, z_{10}, z_{20}, z_{40}, z_{50}, z_{60}, z_{70}$ and let the time prediction network calculate their distance to their successors. Note that for the blue graph, we plot $\Psi_\theta(z_0, z_0), \Psi_\theta(z_0, z_1), \Psi_\theta(z_0, z_2), ..., \Psi_\theta(z_0, z_{72})$. For the orange one we plot $\Psi_\theta(z_{10}, z_{10}), \Psi_\theta(z_{10}, z_{11}), \Psi_\theta(z_{10}, z_{12}), ..., \Psi_\theta(z_{10}, z_{72})$.

Table 2: Mean cumulative reward for Atari (rows 2 to 6 are copied from Kim et al. (2018), all others from the cited papers). This table shows the performance of different exploration-based and baseline methods. Our results are computed as the mean over 20 runs where each run has seen 5M frames.

| Method | Frames | Montezuma | Gravitar | Frostbite |
| --- | --- | --- | --- | --- |
| R2D2 (Kapturowski et al., 2019) | 10000M | 2061 | 15680 | 315456 |
| EX2 (Fu et al., 2017) | 50M | 0 | 550 | 3387 |
| AE-SimHash (Strehl and Littman, 2008) | 50M | 75 | 482 | 5214 |
| ICM (Pathak et al., 2017) | 50M | 161 | 424 | 4465 |
| RND (Burda et al., 2018) | 50M | 377 | 546 | 2227 |
| EMI (Kim et al., 2018) | 50M | 387 | 558 | 7002 |
| LWM (Ermolov and Sebe, 2020) | 50M | 2276 | 1376 | 8409 |
| PPO (Schulman et al., 2017) | 40M | 42 | 737 | 314 |
| **Ours (Time-myopic Go-Explore)** | 5M | 2090 | 3161 | 4476 |
| **Ours (Time-myopic Go-Explore)** | 1M | 695 | 2533 | 3543 |
| Go-Explore (native) | 1M | 2303 | 2130 | 11721 |

the advantage that it can act more greedily, because the representation heuristic is always reliable and therefore can be exploited by acting more risky. Also, native Go-Explore uses the originally recommended down-scaling hyperparameters ($h = 8, w = 11, dp = 8$) for Montezuma's Revenge and the dynamic down-scaling (recompute every 500k frames) only for the other environments (Montezuma's Revenge with dynamic down-scaling performs worse). For native Go-Explore we are using the official implementation. Our time-myopic Go-Explore variant runs on one A100 GPU and needs roughly 8-10 hours for 5M frames depending on the archive size.

Table 3: Comparison of the archive size for Montezuma's Revenge for scores 100, 400, 500, 2500.

| Method | Archive size per score | | | |
| --- | --- | --- | --- | --- |
| | 100 | 400 | 500 | 2500 |
| Native ($h = 8, w = 11, dp = 8$) | 126.2 | 197.8 | 1812.2 | 2838.9 |
| Time-myopic ($L = 20, T_d = 0.55$) | 160.5 | 202.5 | 450.0 | 566.0 |
| Time-myopic ($L = 25, T_d = 0.65$) | 103.8 | 126.2 | 210.7 | 248.3 |

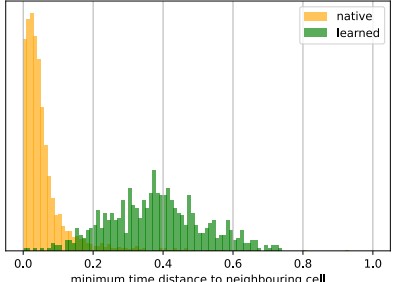

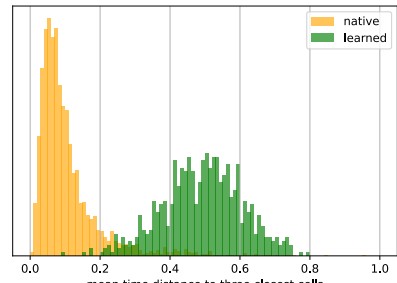

(a) Distance to closest cell neighbor

(b) Mean distance to next three cell neighbors

Figure 4: Time distances between archive cells on Montezuma's Revenge. We generate two archives where the first one (orange) was constructed using Go-Explore down-scaling method ($h = 8$, $w = 11$, $dp = 8$) and the second one (green) by our learned model predictions ($L = 20$, $T_d = 0.55$). When both archives reach the score of 2500 we use a second optimized time prediction network ($L = 20$) and compute all time distances between the cell observations for both archives.

The result shows the practicality of time-myopic Go-Explore since it is better than most of the methods for Montezuma's Revenge and Gravitar while it only has seen 2% of their frames (1M vs 50M frames). It also provides a good performance on Frostbite surpassing PPO, RND, EX2 and ICM. Moreover, time-myopic Go-Explore is able to keep up with the performance of native Go-Explore (using the domain-aligned representation heuristic) and even exceeds it within 1M frames for the environment Gravitar. As far as we know the results of Gravitar for 1M frames are state-of-the-art.

### 6.3 ABLATION STUDY

Next we provide a closer look at the archive properties. First, we compare the difference in the archive size between native Go-Explore and our approach (see Table 3). The table shows the mean number of cells (20 runs) in the archive when the agent reaches a certain game score $[100, 400, 500, 2500]$ in Montezuma's Revenge. The native archive size explodes after reaching a score of 400 while our approach shows a more stable progression. The data also validates smaller archive sizes as a result of increasing the hyperparameters ($L, T_d$). In the Appendix, a figure shows more precisely when and how the prediction model decides to add archive entries. We observe in our experiments a similar pattern for the environment Gravitar which uses the *dynamic* down-scaling heuristic. The native archives agglomerate an enormous amount of archive entries. After 1M frames for 20 runs, the mean size results in 7376 cells while time-myopic Go-Explore holds 381 entries and is achieving a higher score.

Secondly, both methods are evaluated on the similarity within the created archives (see Figure 4). The time distances between all cell observations will provide an interesting similarity measurement. The model predictions in Figure 4 confirm that the native archive yields a lot more similarity than our approach. This leads to inefficient exploration, because based on the cell selection criterion every cell needs to be sufficiently visited, no matter how few environment transitions are between them. Our archives cover the state space with less cells.

### 6.4 LIMITATIONS

Due to the queries to the archive, our algorithm runtime increases currently non-linearly with the number of entries. This makes long runs or runs where the archive grows fast computationally expensive. This happens, because our approach requires to compute time distances to every archive entry in order to evaluate new cell candidates or to update the visit statistics. So, in the future, we will make these queries more efficient to make the method applicable to even larger environments and runs.

## 7 RELATED WORK

Hard exploration problems like the Atari environment Montezuma's Revenge are known for their large state spaces and sparse reward functions. A lot state-of-the-art reinforcement learning algorithms (Schrittwieser et al., 2020; Kapturowski et al., 2019; Schulman et al., 2017) have difficulties in achieving a good performance on these tasks, so several ideas have proposed towards a solution. The literature contains several approaches related to our work that deal with unsupervised- and representation learning combined with intrinsic motivation and exploration.

**Playing hard exploration games by watching YouTube** (Aytar et al., 2018). In this work the authors introduce a neural network architecture in order to learn a categorical time classification between two distinct observations. Their network is used to generate an intrinsic reward signal to facilitate imitation learning of human demonstrations while we are using it to globally model the exploration process.

**Solving sparse reward environments using Go-Explore with learned cell representation** (Bjørsvik, 2021). This approach also extends the Go-Explore (Ecoffet et al., 2021a) method by replacing the representation heuristic with a learned state representation. They employ a Variational Autoencoder (VAE) (Kingma and Welling, 2013) to encode every seen state into a latent vector space. Later on, the encodings are used for a k-means clustering procedure where a cluster center stands for an entry in the archive memory.

**Latent world models for intrinsically motivated exploration** (Ermolov and Sebe, 2020). This paper optimizes a siamese network that clusters temporal imminent states in the embedding space by minimizing the mean squared error between them. In addition, there is the need for an extra structure constraint in order to prevent a collapse of the representation to a constant vector. The resulting encodings are used for the generation of an intrinsic reward signal.

**Episodic curiosity through reachability** (Savinov et al., 2018). The paper introduces a model architecture with a logistic regression capability to differentiate between novel and familiar state pairs where novelty is defined by a minimum time distance of $k$ steps. The complete model (including a siamese encoder and comparison network) is only able to decide whether an encountered pair is novel or not; without the ability to evaluate it on a continuous basis. This estimation is utilized to generate a positive intrinsic reward signal when a policy encounters new states.

**Never give up: learning directed exploration strategies** (Badia et al., 2020). The proposed episodic novelty module starts empty and fills itself with state encodings when the agent interacts with the environment. Every state gets evaluated by a k-nearest neighbor criterion with the similarity measure of the Dirac delta kernel to compute the rewards with respect to the similarity distance. The goal is to insert as many novel states as possible which then facilitate exploration.

## 8 CONCLUSION

To add flexibility to and possibly improve Go-Explore we studied how its representation heuristic can be replaced by a time-predicting neural network. Experiments show that the new state representation is able to track the global exploration effort and moreover recognizes ongoing progress for this task. In comparison to native Go-Explore, our method can reduce the archive size and covers the state space with fewer cells. Applied to hard exploration environments, such as Montezuma's Revenge we observe good performance compared to previous exploration-based methods while using much fewer observations, even creating better results than Go-Explore on the game Gravitar. Overall, however our learned representation is not able to compete with native Go-Explore in terms of sample efficiency.

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

# A   APPENDIX - SECTION 1

Table 4: Hyperparameters for the experiments reported in the hard exploration table.

| Hyperparameter | Symbol | Value (MontezumaRevenge, Gravitar, Frostbite) |
|---|---|---|
| Learning rate | $lr$ | $1e^{-4}$ |
| Batch size | $bs$ | 64 |
| Time window | $L$ | $(20, 25, 25)$ |
| Distance threshold | $T_d$ | $(0.6, 0.75, 0.75)$ |
| Visit threshold | $T_v$ | 0.65 |
| Proxy cell interval | $[T_{\text{p-low}}, T_{\text{p-high}}]$ | $[0.45, 0.75]$ |
| Exploration steps | $t$ | 40 |
| action repetition mean | $\mu$ | 4 |

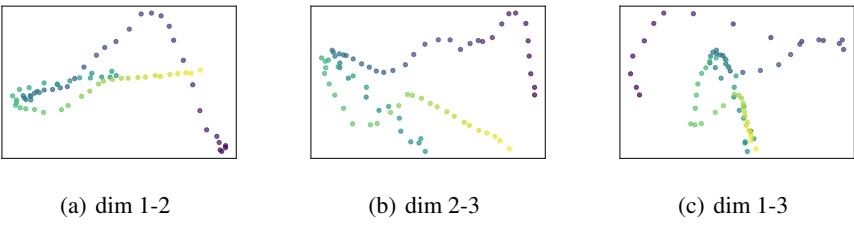

(a) dim 1-2            (b) dim 2-3            (c) dim 1-3

Figure 5: Principal Component Analysis for the trajectory visualization.

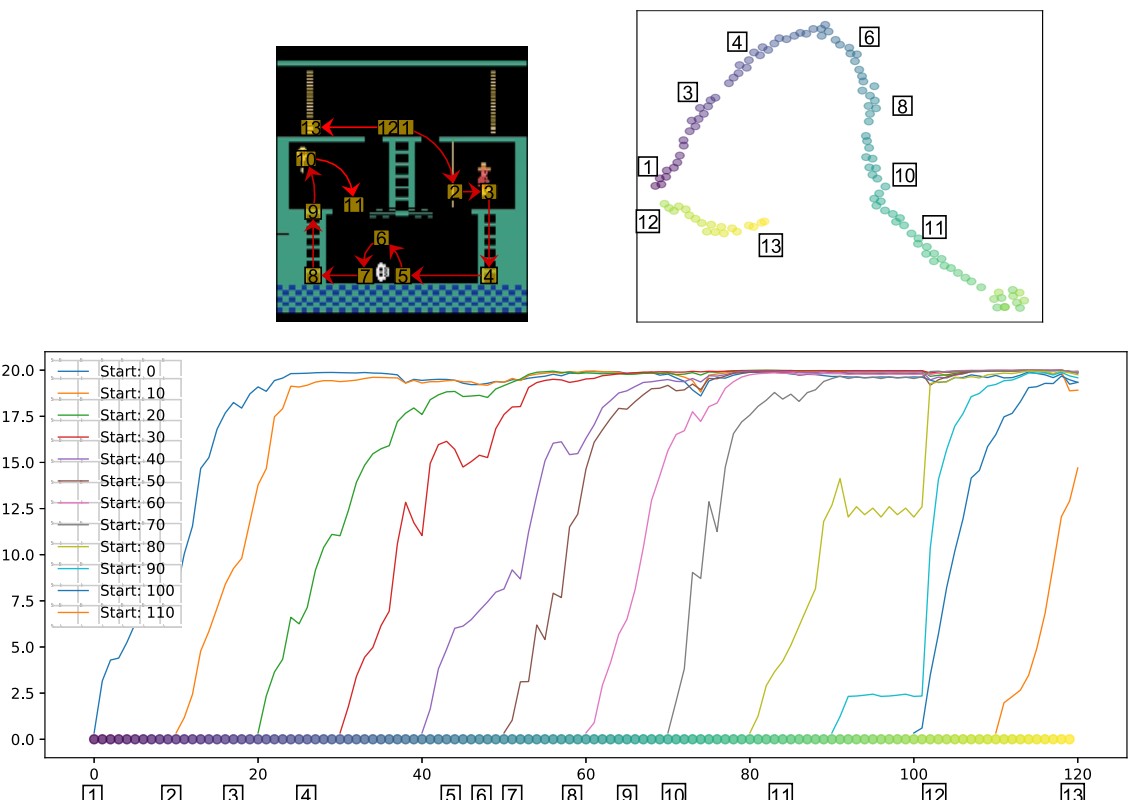

Figure 6: Sophisticated model and its capabilities on a longer trajectory. The shown model is trained on data that surpasses the shown trajectory (top-left). Again, we can see the good encoding property and an improved time prediction skill. The predicted times around the timesteps 90-100 or in the image at the Box 11 are accurate. At that point the agent dies and the environment generates repeating frames (two-image sprites) for around 10 frames. This prediction behavior happens, because we remove temporal ambiguity between state-pairs and try to calculate the shortest distance for it. The event can also be seen in the t-SNE visualization, when the light-green points start to form a cluster.

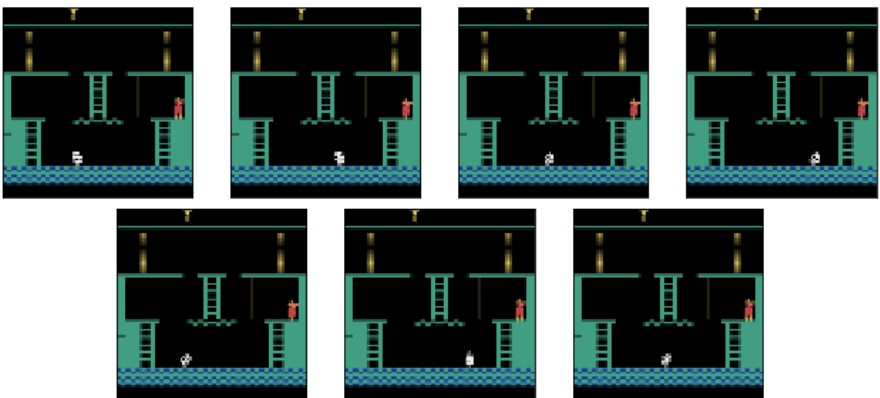

Figure 7: Creation of archive entries. We manually looked through the cell observations of an archive and searched for similarity. This figure shows *all* cell observations where the agent stands at the same position and already collected the key. We can see that the time prediction network does not allow duplicates in the archive and keeps a reasonable distance between the observations where the white skull is changing positions. Moreover the archive holds no observation where the agent just slightly moved in these situations.

---

**Algorithm 1** Go-Explore with a learned state representation

---

Initialize archive, dataset, agent, network
**for** iteration $= 1, 2, ...$ **do**
    Let the agent act $t$ timesteps in the environment starting from the selected and proxy cells
    Collect data and optimize $L^{\text{time}}$ w.r.t. $\theta$
    Recompute all archive cell representations: $z_C = \Phi_\theta(\bar{o}_C)$
    **for each** trajectory $= 1, 2, ..., N$ **do**
        Transfer all observations $\bar{o}_1, ..., \bar{o}_t$ into the latent representation $z_1, ..., z_t$
        Compute all necessary time distances $\Psi_\theta(z_c, z_{1,...,t})$
        Apply archive insertion criterion to a candidate w.r.t. threshold $T_d$
        Increase cell visits w.r.t. threshold $T_v$
        Collect some proxy cells w.r.t. threshold $[T_{\text{p-low}}, T_{\text{p-high}}]$
        **if** candidate is accepted **then**
            Add candidate to archive
        **end if**
    **end for**
**end for**

---

