# OpenReview forum: "Time-Myopic Go-Explore: Learning A State Representation for the Go-Explore Paradigm"
_ICLR.cc/2023/Conference — Submitted to ICLR 2023_

### Official Review · Reviewer_ALFv · 2022-10-18

**Confidence:** 3
**Correctness:** 3
**Technical Novelty And Significance:** 2
**Empirical Novelty And Significance:** 2
**Recommendation:** 3

**Clarity, Quality, Novelty And Reproducibility:**

The paper is well written but several details such as the local dataset and explanation of worse results are missing.

**Strength And Weaknesses:**

**[Strength]**

1. The proposed method that considers temporal difference for storing the observations for Go-Explore is straightforward and seems effective in terms of reducing the temporal redundancy and diversifying the archived observations.

2. The paper is generally well-written and easy to understand.

3. The visualization from an example trajectory (Figure 3) makes a better understanding of time prediction.

**[Weakness]**

1. According to Table 3, L and Td hugely affect the size of the archive. The sensitivity analysis of the hyperparameters is needed.

2. The description of Eq. (5) is insufficient. How to define the score weight term, especially, how to choose 0.075?

3. The performance in Frostbite is notoriously worse compared to vanilla Go-Explore. Moreover, considering the performance in Montezuma's revenge, it seems that it requires 5 times more frames to achieve a slightly worse performance than Go-Explore (sample inefficient). The authors did not deal with these performance gaps.

4. In Section 5.3, the authors mention that they added “some small local dataset for each cell” but it is not well described.

5. It will be better if the authors can present the comparison with Go-Explore in many different Atari games and other tasks, and analyze when the proposed method is effective and when it is not.

**[minor]**

1. It is good to unify the citation for Go-Explore into one (Nature version).

2. In Table 1, it is better to change ‘Td > 13’ to ‘Td > 0.65’ for consistency.


**Summary Of The Paper:**

The paper proposes a time-myopic state representation that estimates the temporal difference between two observations. The time prediction is trained by MSE loss with the normalized temporal difference between two observations from the sampled trajectories. The authors apply this time prediction to Go-Explore (Ecoffet et al. 2021) for achiving more temporally distant (abstracted) observations and reducing the number of elements in the archive. Consequently, the proposed Time-Myoptic Go-Explore achieves better performance than vanilla Go-Explore in Gravitar although it has much worse performance in Frostbite.


**Summary Of The Review:**

Although considering temporal distance for storing cells in Go-Explore is interesting and has reasonable improvement in terms of memory efficiency, the performance is generally degraded and requires more frames (sample-inefficient) compared to vanilla Go-Explore. Moreover, the authors only conducted experiments on three environments, and the results are not consistent. Consequently, I gave a reject in the initial review.

**Update**
After reading the authors' responses and other reviews, several concerns are still unsolved and the authors did not respond to the concerns. Hence, I maintain the rating.

---

> ### Author Response · Authors · 2022-11-17
> **Reviewer ALFv**
>
> We thank the reviewer for their detailed reviews.
>
> > The description of Eq. (5) is insufficient. How to define the score weight term, especially, how to choose 0.075?
> The description of the Cscore was missing and has been added to the revised version.
>
> You are right 0.075 is a hyperparameter.  However, in our experience the exact value was quite uncritical for the performance of the method.
>
> > The performance in Frostbite is notoriously worse compared to vanilla Go-Explore. Moreover, considering the performance in Montezuma's revenge, it seems that it requires 5 times more frames to achieve a slightly worse performance than Go-Explore (sample inefficient). The authors did not deal with these performance gaps.
>
> We note that our method achieves after 5M frames much better results than other established methods after 50M frames.  We do discuss the comparison with native Go-Explore in detail in Section 6.2.
>
> > In Section 5.3, the authors mention that they added “some small local dataset for each cell” but it is not well described.
>
> We revised that section to better explain our motivation.
>
> > It will be better if the authors can present the comparison with Go-Explore in many different Atari games and other tasks, and analyze when the proposed method is effective and when it is not.
>
> Of course, more experiments are always good.  However, note that the computational resources are limited for most research groups.
>
> > It is good to unify the citation for Go-Explore into one (Nature version).
>
> This is not possible because the earlier Go-Explore version includes an experiment that is not shown in the Nature version.
>
> > In Table 1, it is better to change ‘Td > 13’ to ‘Td > 0.65’ for consistency.
>
> Thank you for pointing this out.  We change it to '20*Td>13'.
>
> > The paper is well written but several details such as the local dataset and explanation of worse results are missing.
>
> The comparison with native Go-Explore is discussed in Section 6.2 and we added details for the local dataset in the revised version.
>
> > ... and the results are not consistent.
>
> We are not sure what you mean by "not consistent".  Our approach shows almost on par results with Go-Explore (even better on Gravitar) for only 1M frames and excellent results after 5M frames compared to other exploration-based methods that use 50M frames.

---

> > ### Comment · Reviewer_ALFv · 2022-11-17
> > **Still have concerns**
> >
> > Thank you for responding to my concerns and changing the manuscript.
> >
> > > You are right 0.075 is a hyperparameter. However, in our experience, the exact value was quite uncritical for the performance of the method.
> >
> > The authors mention $\alpha$ is `quite uncritical`. However, I cannot find any sensitivity analysis about it in the paper. Supporting results should exist to argue that the proposed method is robust to various $\alpha$.
> >
> >
> > > We note that our method achieves after 5M frames much better results than other established methods after 50M frames. We do discuss the comparison with native Go-Explore in detail in Section 6.2.
> >
> > In Table, 2, Go-Explore also uses only 1M frames to achieve 2303 in Montezuma's Revenge and 11721 Frostbite while the proposed method achieves 2090, and 4476 in Montezuma's Revenge and Frostbite, respectively in 5M settings.
> >
> > p.s. In general, all figures and tables should be mentioned in the main text. I recommend mentioning Table 2 somewhere.
> >
> > > Of course, more experiments are always good. However, note that the computational resources are limited for most research groups.
> >
> > I am confused by this statement. In Section 6.2, the authors explained that it only takes 8-10 hours for 5M frames with a single A100 GPU. This means that even with a much worse GPU will not take several days.
> >
> > I understand that running all Atari games requires lots of resources but I believe adding a few more environments to present that the proposed method actually outperforms Go-Explore is still helpful and not a huge burden to authors.
> >
> >
> > > This is not possible because the earlier Go-Explore version includes an experiment that is not shown in the Nature version.
> >
> > If I understand correctly, all experiments in the arXiv version are included in the Nature version (including supplementary information). If not, it is good to use the arXiv version solely unless some part in the Nature version is not listed in the arXiv version.
> >
> >
> > > We are not sure what you mean by "not consistent". Our approach shows almost on par results with Go-Explore (even better on Gravitar) for only 1M frames and excellent results after 5M frames compared to other exploration-based methods that use 50M frames.
> >
> > I intended that the results do not consistently outperform Go-Explore in the same amount of frames (Table 2) and only it outperforms in Gravitar. I apologize for the confusion.

---

### Official Review · Reviewer_7Zd7 · 2022-10-24

**Confidence:** 3
**Correctness:** 1
**Technical Novelty And Significance:** 3
**Empirical Novelty And Significance:** 3
**Recommendation:** 3

**Clarity, Quality, Novelty And Reproducibility:**

This paper can be greatly improved by providing a more self-contained description of the problems and explicitly state the issues and contributions of the work, such as how does the time predictive semantics help the detachment part? These results have not been provided.
Comments/Suggestions on paper improvement:
Why would myopic prediction be good ? isn’t this going against the idea of go explore?
How is the policy affecting the time predictions? It is unclear from the text.
Where are the weights  in eq. 5 being used? How are Cscore and Cvisit being counted, in particular Cscore? (a more self-contained description would benefit clarity of the paper)
What do the authors mean by extrapolation in 6.1? Of what?

Limitations of the current approach have not been addressed either, I would suggest addressing the difficulties of estimating time predictive representations and how the current approach limits exploration, is spatial information still captured? To what extent?.


**Strength And Weaknesses:**

Strengths:
This paper focuses on a very relevant problem in exploration of state representation and introduces a novel and original way of measuring novelty.
Weaknesses:
The paper lacks scientific rigour. The authors show experiments in table 2 with different frame counts making it very hard to compare among results. And even for the 1M frame budget the Go-Explore native seems to perform better. It is also unclear how precise the time predictive network is for each k, under each game. Is the time prediction correct on new seeds? One of the main arguments for the inadequacy of Go-Explore is that the overlapping conflicts lead to states being abandoned and not further pursued for exploration, how much conflicts is taking place quantitatively and does this lead to a significant performance drop, with the results presented so far it is still unclear this is true and time prediction mitigates this.
The paper is also not very clear, for example for the cell selection criteria Cvisits and Cscores is not defined, not even in the text. (see questions below for more)
The motivation for starting in proxy cell states is not clearly explained as well, how and where would this help/impact.?


**Summary Of The Paper:**

This paper proposes new learned state representations that can improve exploration by introducing time predictive semantics into the state representation. The authors propose a new notion of novelty based on time prediction representations and use this to create a new insertion criterion, a new cell count strategy and a new restart strategy for exploration in the settings of Go-Explore.

**Summary Of The Review:**

This paper introduces time predictive state representations and uses them as notion of novelty using in the go-explore context to store states and as different visitation counts, and new cell restarts, the paper provides an interesting perspective but lacks in clarity and scientific rigour. Further experiments would be needed to support the claims and provide a scientific understanding of the contributions proposed.

---

> ### Author Response · Authors · 2022-11-17
> **Reviewer 7Zd7**
>
> We thank the reviewer for their detailed comments.
>
> > Weaknesses: The paper lacks scientific rigour. The authors show experiments in table 2 with different frame counts making it very hard to compare among results.
>
> We do not agree: if a method running on 50M frames reaches a lower score than our method running for 5M frames, it is very clear that our method is much better.  So, the comparison is very clear.
>
> > And even for the 1M frame budget the Go-Explore native seems to perform better.
>
> In the 1M regime Go-Explore is better in two games and we are better in one game.  As we discussed in Section 6.2 the reasons for the outcome.
>
> > It is also unclear how precise the time predictive network is for each k, under each game. Is the time prediction correct on new seeds?
>
> Please do note that Table 2 shows the results of our method after 20 randomly initialized (with different seeds) runs. If the time prediction quality is bad we would not see the reported performance. The archive would be flooded with (similar) states, which is not the case (see ablation and Table 3) for Montezuma and Gravitar.
>
> > One of the main arguments for the inadequacy of Go-Explore is that the overlapping conflicts lead to states being abandoned and not further pursued for exploration, how much conflicts is taking place quantitatively and does this lead to a significant performance drop, with the results presented so far it is still unclear this is true and time prediction mitigates this.
>
> Your question about the abandoned archive states is good.  However, in the original implementation of Go-Explore this information is very difficult to extract.
>
> > The paper is also not very clear, for example for the cell selection criteria Cvisits and Cscores is not defined, not even in the text. (see questions below for more)
>
> The C-visits are from the original paper.  The description of the C-scores was missing and has been added to the revised version.
>
> > The motivation for starting in proxy cell states is not clearly explained as well, how and where would this help/impact.?
>
> The basic motivation for the proxy cells is to increase training data variability.  We didn't include an ablation but in our experience there was a clear advantage.
>
> > This paper can be greatly improved by providing a more self-contained description of the problems and explicitly state the issues and contributions of the work, such as how does the time predictive semantics help the detachment part?
>
> Note that our approach never removes information from the archive, so there is no detachment problem anymore.  We added a sentence on this fact to Section 4.1.
>
> > Why would myopic prediction be good ? isn’t this going against the idea of go explore?
>
> No, it is not against the idea of go explore, since we are still using the archive and consider local distances everywhere and apply this evaluation globally over the whole archive.
>
> > How is the policy affecting the time predictions? It is unclear from the text.
>
> This point is discussed in Section 3.4.
>
> > Where are the weights in eq. 5 being used? How are Cscore and Cvisit being counted, in particular Cscore? (a more self-contained description would benefit clarity of the paper)
>
> We add an explanation of Cvisits Cscore to the revised version.
>
> > Limitations of the current approach have not been addressed either ...
>
> We discuss the limitations of our method in Section 6.4.

---

> > ### Comment · Reviewer_7Zd7 · 2022-12-12
> > **Position unchanged**
> >
> > I would like to thank the authors for their response and the effort put into improving the clarity of the paper, however most of my concerns still remain, hence I will keep my score.

---

### Official Review · Reviewer_RwfW · 2022-10-25

**Confidence:** 4
**Correctness:** 2
**Technical Novelty And Significance:** 2
**Empirical Novelty And Significance:** 2
**Recommendation:** 3

**Clarity, Quality, Novelty And Reproducibility:**

- Clarity: The paper is generally well-written and quite clear. Some details could be improved, e.g. in section 4.1, “the agent samples trajectories from c_1” - is this done by resetting the environment state, or by getting the agent to return?
- Quality: The explanation and inspection of the proposed mechanism are well done. The main quality issue is the experimental evaluation, which is seriously lacking in width (only three Atari games) and depth (only the initial phases of those games). Questions about the detailed choices of the time distance modeling are a bit moot because of the limited evaluation, but it would have been interesting to discuss things like the time scale introduced through the cutoff L, and the fact that only the difference between the states is fed into the time predictor - wouldn’t there be settings where the constant part is useful when trying to predict the time difference between the states?
- Novelty: The notion of modelling a difference between two states to obtain an exploration signal is not new, but the specific version here of modelling time difference and using that model output directly as a novelty signal, in Go-Explore, is new as far as I’m aware.
- Reproducibility: no concerns.


**Strength And Weaknesses:**

- Major weakness: the paper only contains results on Montezuma’s Revenge, Gravitar, and Frostbite, over very few frames. There is no evidence in the paper that the method generalizes beyond this regime.
- Strength: the idea to use time difference predictions as a basis for novelty estimations is in principle sound.
- Strength: the paper is well-written and easy to read.
- Strength: the proposed novelty estimator is studied in some detail in the paper, and shown to produce a good distribution of states entered into memory.


**Summary Of The Paper:**

The paper proposes a learned novelty estimator, based on the temporal distance between states, to be used in the Go-Explore architecture as a basis for memory writing decisions. The proposed method is evaluated in the low-data regime on three hard exploration Atari games, where most baselines are outperformed. The effects of using the proposed novelty estimator are investigated in some detail, and compare favorably to the native Go-Explore one.

**Summary Of The Review:**

The idea behind the paper, of modeling time distances for exploration, is a good one in principle. However, the lack of any evidence that the proposed method generalizes is a major gap that renders the paper unsuitable for publication. I recommend rejection.

---

> ### Author Response · Authors · 2022-11-17
> **Reviewer RwfW**
>
> We thank the reviewer for their detailed assessment.
>
> > Clarity: The paper is generally well-written and quite clear. Some details could be improved, e.g. in section 4.1, “the agent samples trajectories from c_1” - is this done by resetting the environment state, or by getting the agent to return?
>
> Currently we are restoring the environment state, but there are no limitations, because the author of Go-Explore showed that it is possible to learn a return policy.
>
> > Quality: The explanation and inspection of the proposed mechanism are well done. The main quality issue is the experimental evaluation, which is seriously lacking in width (only three Atari games) and depth (only the initial phases of those games).
>
> Note that most related methods are not even going beyond the initial phase of the game after 50M steps.
>
> > Questions about the detailed choices of the time distance modeling are a bit moot because of the limited evaluation, but it would have been interesting to discuss things like the time scale introduced through the cutoff L, and the fact that only the difference between the states is fed into the time predictor - wouldn’t there be settings where the constant part is useful when trying to predict the time difference between the states?
>
> We focus on the time difference between states because that corresponds to some distance measure in the state space to decide whether a given state made progress compared to the other states in the archive.
>
> > Novelty: The notion of modeling a difference between two states to obtain an exploration signal is not new, but the specific version here of modeling time difference and using that model output directly as a novelty signal, in Go-Explore, is new as far as I’m aware.
>
> Of course, we are interested in more references to other works that use the difference between two states as an exploration signal, so we would be happy if you add them to your review, then we can include them into our paper.

---

> > ### Comment · Reviewer_RwfW · 2022-12-08
> > **Position unchanged**
> >
> > Apologies for my late response.
> >
> > I thank the authors for their response to my review. My main concern, that the paper does not demonstrate the generalizability of the proposed method, remains, as the authors have not addressed it. Hence my score is unchanged.

---

### Official Review · Reviewer_Ze5a · 2022-10-26

**Confidence:** 4
**Correctness:** 2
**Technical Novelty And Significance:** 2
**Empirical Novelty And Significance:** 2
**Recommendation:** 3

**Clarity, Quality, Novelty And Reproducibility:**

Clarity:
- Low, see my detailed comments above.

Quality:
- Medium. Results are ok but it would be nice to have additional results for different sample complexity budgets and the ablation described above.

Novelty:
- Medium. The algorithm proposes a novel modification to an existing algorithm.

Reproducibility:
- High. The authors promise to open source their code.

**Strength And Weaknesses:**

## Strengths
- Although the original Go-Explore algorithm achieved impressive performance, it used several cheats like using handcrafted state abstractions (based on downsampling images), resetting the environment (or equivalently, exploiting the determinism of the simulator) which are not generally applicable. It would be great to have a version of Go-Explore which keeps the good qualities and performance but does away with these hacks, which this paper tries to make progress towards.

## Weaknesses
- As far as i can tell, this paper still assumes a deterministic simulator or the ability to reset to previously visited states, which is still a very strong assumption. So the version of Go-Explore it proposes is still not generally applicable.
- The paper is not very clearly written - see my comments below
- The results are decent, but it is currently unclear how the proposed algorithm scales with more data (only performance for 1M and 5M are reported). Please report performance at least until 50M so we can compare to the other model-free approaches.

Detailed comments:
- Sections 1 and 2 are confusing. It was not clear to me until Section 4.1 what the point of predicting the number of timesteps k connecting two different states was. Section 2 is very confusing, with a lot of vague statements such as "The conflict solver must resolve illogical and non-intuitive conflicts" or "The down-scaling method is compressing the image information without considering its semantic content". Couldn't this be said about many different unsupervised learning methods? In the first paragraph, it seems to be saying that state abstraction is a bad thing, and that the fact that many states are grouped to the same representation is a bad thing. Can't this also be a good thing in that it filters out irrelevant details? It all depends on _how_ the grouping is done. I would suggest providing a detailed motivating example (from Montezuma's Revenge perhaps) that illustrates all of these claimed problems with Go-Explore - having a concrete example illustrating these statements would make this part much clearer.

- It's not clear to me whether the timestep predictor is even necessary. Since you have access to the actual time step, and you are restarting exploration from different states in the archive, you should at least have the number of timesteps between a given state and the state the trajectory was restarted from. What happens if you use this as your archive insertion criterion? It might not work but should be a baseline to justify the need for the learned predictor. A simple baseline could be: record the timestep t of each state, then use the difference between the true timesteps for the archive insertion criterion.



There are a number of typos/grammar mistakes - please proofread carefully. A few examples:
- Abstract: "clusters temporal close states" -> "clusters temporally close states"
- Section 3.2: "a pair of observation encodings is allow us to estimate" -> "a pair of observation encodings allows us to estimate"
- Section 4.1: "An short example" -> "A short example"
- Section 4.1: "is shown Table 1" -> "is shown in Table 1"
- Section 6.2: "shows the practicability" -> "shows the practicality"
- Section 6.2: "It is also providing a good performance" -> "It also provides a good performance"


**Summary Of The Paper:**

This paper proposes a modification of the Go-Explore algorithm to remove the need for hardcoded state representations (which are used in the original version of the algorithm). Specifically, this work proposes to learn an embedding using an auxiliary loss based on predicting the number of timesteps connecting two observations. This timestep prediction function is also used to populate Go-Explores's goal archive: specifically, states are only added to the archive if they are at least a certain distance from all other states in the archive. This is to ensure that new states/goals are only added if they expand the frontier of explored states.

The proposed method is evaluated on 3 hard exploration Atari games: Montezuma's Revenge, Gravitar and Frostbite. It doesn't quite match the original Go-Explore, but it does decently well compared to other baselines in terms of performance and sample complexity.

**Summary Of The Review:**

Overall, I think that the paper's goal of making Go-Explore more general is a good one, and there are some interesting ideas in this paper. However, I believe it needs a good amount of work before it's ready for publication, specifically in terms of its clarity and its experimental results.

---

> ### Author Response · Authors · 2022-11-17
> **Reviewer Ze5a**
>
> We thank the reviewer for their detailed reviews.
>
> > As far as i can tell, this paper still assumes a deterministic simulator or the ability to reset to previously visited states, which is still a very strong assumption. So the version of Go-Explore it proposes is still not generally applicable.
>
> The authors of the original Go-Explore paper demonstrated the ability to return to a state (stored additionally in the archive) with a learned policy.  The same trick can be applied to our work.
>
> > The paper is not very clearly written - see my comments below (Details section)
>
> Thank you for the detailed comments which helped us to improve the wording and clarity.
>
> > The results are decent, but it is currently unclear how the proposed algorithm scales with more data (only performance for 1M and 5M are reported). Please report performance at least until 50M so we can compare to the other model-free approaches.
>
> We do observe a great improvement from 1M to 5M, so in that regime it scales nicely.  However, reasonable experiments with 50M are beyond our computational resources.
>
> > Sections 1 and 2 are confusing. It was not clear to me until Section 4.1 what the point of predicting the number of timesteps k connecting two different states was. Section 2 is very confusing, with a lot of vague statements such as "The conflict solver must resolve illogical and non-intuitive conflicts" or "The down-scaling method is compressing the image information without considering its semantic content". Couldn't this be said about many different unsupervised learning methods?
>
> Yes, that is the case for many methods.  The semantic content might help predicting the correct time difference.  We tried to edit Sections 1 and 2 to make them more clear.
>
> > In the first paragraph, it seems to be saying that state abstraction is a bad thing, and that the fact that many states are grouped to the same representation is a bad thing. Can't this also be a good thing in that it filters out irrelevant details? It all depends on how the grouping is done.
>
> You are right, that it depends on how the grouping is done.  That is why we are proposing to learn the time distance which implicitly determines the grouping.
>
> > It's not clear to me whether the timestep predictor is even necessary. Since you have access to the actual time step, and you are restarting exploration from different states in the archive, you should at least have the number of timesteps between a given state and the state the trajectory was restarted from. What happens if you use this as your archive insertion criterion? It might not work but should be a baseline to justify the need for the learned predictor. A simple baseline could be: record the timestep t of each state, then use the difference between the true timesteps for the archive insertion criterion
>
> The problem with just measuring the time passed from the origin is not sufficient, because this strategy would ignore the time difference to all other archive entries.  The state might be far away from the restore point, but we would have no information about the distance to the other restore states in the archive. We did try to add states randomly, which did not work at all and is somewhat similar to your proposal.

---

> > ### Comment · Reviewer_Ze5a · 2022-12-09
> > **score remains the same**
> >
> > Thank you for the response. While the authors have answered some of my concerns in their rebuttal, they have not run the experiments required to truly address these concerns (see below). Therefore, my score is unchanged.
> >
> > - _The authors of the original Go-Explore paper demonstrated the ability to return to a state (stored additionally in the archive) with a learned policy. The same trick can be applied to our work._
> >
> > While this may be possible in principle, this experiment should be included in the paper to show that this works in practice for the proposed method.
> >
> > - _We do observe a great improvement from 1M to 5M, so in that regime it scales nicely. However, reasonable experiments with 50M are beyond our computational resources._
> >
> > In Section 6.2 it says that training for 5M frames takes 8-10 hours on a single GPU. Training for 50M steps could be run in 3-4 days, which I don't think is unreasonable.
> >
> > - _The problem with just measuring the time passed from the origin is not sufficient, because this strategy would ignore the time difference to all other archive entries. The state might be far away from the restore point, but we would have no information about the distance to the other restore states in the archive. We did try to add states randomly, which did not work at all and is somewhat similar to your proposal._
> >
> > It would have been nice to include an experiment showing this.

---

### Decision · Program_Chairs · 2023-01-20

**Decision:**

Reject

**Justification For Why Not Higher Score:**

Reviewers pointed out numerous major concerns about the technical quality of this approach, quality of the presentation, and missing experiments, all of which preclude acceptance. After the rebuttal period, they were still not convinced, and hence I recommend reject.

**Justification For Why Not Lower Score:**

N/A

**Metareview: Summary, Strengths And Weaknesses:**

This paper extends the Go-Explore method by learning an embedding rather than hardcoding state representations, which can be used for novelty-guided exploration. They evaluated on Montezuma’s Revenge, Gravitar, and Frostbite, yielding results that are comparable to the original Go-Explore approach, but without a need for handcrafted representations.

Reviewers felt that this was a straightforward yet still novel extension of the original algorithm, commended the paper for being overall well-written and easy to read (although many clarity issues were also pointed out), and appreciated the analyses and visualization of trajectories.

However, there were some major concerns regarding just how general purpose this approach really is, given that it’s only tested in 3 Atari environments and still requires access to a deterministic simulator in order to reset to previous states. Some thought the paper needed more care and proofreading, and others felt that the scientific rigor could be improved (ie sensitivity analysis, controlling for the number of frames, and testing on more tasks). Two reviewers also pointed out in their follow-up to the authors' rebuttal that the additional experiments requested would not be overly burdensome, and all reviewers who did follow-up indicated that they didn’t find their concerns satisfactorily addressed.

Hence, I believe this paper isn’t ready for publication at ICLR in its current form, but hope that reviewers’ comments are helpful for future versions of this work.